# REDUCE, REUSE, RECYCLE: SELECTIVE REINCARNATION IN MULTI-AGENT REINFORCEMENT LEARNING

## ABSTRACT

'Reincarnation' in reinforcement learning has been proposed as a formalisation of reusing prior computation from past experiments when training an agent in an environment. In this paper, we present a brief foray into the paradigm of reincarnation in the *multi-agent* (MA) context. We consider the case where only some agents are reincarnated, whereas the others are trained from scratch – *selective* reincarnation. In the fully-cooperative MA setting with heterogeneous agents, we demonstrate that selective reincarnation can lead to higher returns than training fully from scratch, and faster convergence than training with full reincarnation. However, the choice of *which* agents to reincarnate in a heterogeneous system is vitally important to the outcome of the training – in fact, a poor choice can lead to considerably worse results than the alternatives. We argue that a rich field of work exists here, and we hope that our effort catalyses further energy in bringing the topic of reincarnation to the multi-agent realm.

## 1 INTRODUCTION

Reinforcement Learning (RL) is a field that has existed for many years, but has recently seen an explosion of interest and research efforts. Since the incorporation of deep neural networks into the paradigm (Mnih et al., 2013), the community has witnessed success in a wide array of tasks, many of which previously seemed intractable (Silver et al., 2016). A commonly-cited feat is achieving superhuman performance in various games, both classical (Schrittwieser et al., 2020) and modern (Berner et al., 2019; Wurman et al., 2022). Such games can represent situations which are high-dimensional, combinatorially complex, and non-linear, and thus demonstrate the sophistication of the RL approach to sequential decision making. Even with the successes of single-agent RL, many real-world settings are inherently multi-agent, where multiple diverse agents act together in a shared environment. The success of Multi-Agent Reinforcement Learning (MARL) has been similarly captivating in this context, with demonstrations of emergence of high-level concepts such as coordination and teamwork (Samvelyan et al., 2019), and even trade (Johanson et al., 2022).

Despite these victories, the discipline of RL still faces a series of fierce challenges when applied to *real-world* situations, not least the intense computation often required for training (Agarwal et al., 2022). The multi-agent case, though highly applicable to the real world, is plagued further by problems of non-stationarity (Papoudakis et al., 2019), partial observability (Papoudakis et al., 2021), and the 'curse of dimensionality' (Du & Ding, 2021). We postulate that RL, and MARL specifically, is a powerful tool to help us model, understand, and solve complex processes and phenomena. First, though, it is clear that these challenges must be mitigated.

Progress is being made in this regard, across a host of research strategies such as transfer learning (Zhu et al., 2020), ad hoc teamwork (Stone et al., 2010), and zero-shot coordination (Hu et al., 2020). Another crucial effort is to leverage prior computation, to avoid the unnecessary duplication of work. In a typical RL research project, an algorithm is trained *tabula rasa* – that is, without prior experience or encoded knowledge. Sometimes, such an approach is desirable: for example, it was the express intention of Silver et al. (2017) to train their AlphaZero agent *tabula rasa*, for the sake of learning to play Go without learning from human data. However, in many practical settings, training from scratch every time is slow, expensive, and also *unnecessary*. For example, we may want to iterate on a problem or test out a new strategy, and do so quickly, without starting over in each case.

In this vein, Agarwal et al. (2022) have recently proposed a formalisation of a research paradigm entitled 'Reincarnating RL,' where previous computation is reused in future work. These authors argue that large, real-world RL systems already take this approach, out of necessity, but in a way that is often ad hoc and informal. Through the creation of a reincarnation framework, not only does a researcher gain benefits in their own experiments, it further allows the field itself to be democratised – enabling the sharing of checkpoints, model weights, offline datasets, etc., to accelerate development. This dimension is particularly salient for low-resourced researchers, who can piggyback off the computing power available to large research labs. Reincarnation is certainly not a panacea for the real-world challenges of RL, but it does provide a springboard both for novel ideas and for new researchers to enter the field. We resonate with this call, and wish to motivate similarly for reincarnation in the MARL context.

To catalyse the excitement for this paradigm, we focus in this paper on a particular aspect of reincarnation that may be useful in MARL: *selective* reincarnation. To illustrate where such a situation is applicable, consider an example of controlling a large, complex industrial plant, consisting of an assortment of *heterogeneous* agents. Notice that this scenario is in the realm of real-world problems. Suppose we are training our system using a MARL algorithm with a decentralised controller, but this training is computationally expensive, on the order of days-long. Conceivably, we may notice that some agents in our system learn competently – perhaps their task is simpler, or the algorithmic design suits their intended behaviour; call these the X agents. Other agents might not fare as well and we would like to train them from scratch; call these the Y agents. We wish to find new strategies for the Y agents: maybe we ought to test a new exploration routine, a novel neural architecture, or a different framing of the problem. Instead of retraining the entire system from scratch after each change in our Y agent strategy, we wonder if we can selectively reincarnate the already-performant X agents and thereby enable faster training times or higher performance for the Y agents.

In this paper, we make three contributions. Firstly, we hope to usher in this nascent paradigm of reincarnation to MARL, where it is vitally needed. The underlying philosophy of leveraging prior computation already exists in the MARL setting (e.g. Kono et al. (2014)), but we aim to begin formalising the field, as done by Agarwal et al. (2022) for the single-agent case. Specifically, we formalise the concept of *selective* reincarnation. Secondly, we demonstrate a few interesting phenomena that arise during a preliminary selectively-reincarnated MARL experiment. We find that, with certain agent subsets, selective reincarnation can yield higher returns than training from scratch, and faster convergence than training with full reincarnation. Interestingly, though, other subsets result in the opposite: markedly worse returns. We present these results as a doorway to a rich landscape of ideas and open questions. Thirdly, we offer a codebase* as a framework for selective reincarnation in MARL, from which other researchers can build.

## 2 PRELIMINARIES

### 2.1 MULTI-AGENT REINFORCEMENT LEARNING

There are many different formulations for MARL tasks including competitive, cooperative and mixed settings. The focus of this work is on the cooperative setting. Fully cooperative MARL with shared rewards can be formulated as a *decentralised partially observable Markov decision process* (Dec-POMDP) (Bernstein et al., 2002). A Dec-POMDP consists of a tuple $\mathcal{M} = (\mathcal{N}, \mathcal{S}, \{\mathcal{A}^i\}, \{\mathcal{O}^i\}, P, E, \rho_0, r, \gamma)$, where $\mathcal{N} \equiv \{1, \ldots, n\}$ is the set of $n$ agents in the system and $s \in \mathcal{S}$ describes the true state of the system. The initial state distribution is given by $\rho_0$. However, each agent $i \in \mathcal{N}$ receives only partial information from the environment in the form of observations given according to an emission function $E(o_t|s_t, i)$. At each timestep $t$, each agent receives a local observation $o_t^i$ and chooses an action $a_t^i \in \mathcal{A}^i$ to form a joint action $\mathbf{a}_t \in \mathcal{A} \equiv \prod_i^N \mathcal{A}^i$. Typically under partial observability, each agent maintains an observation history $o_{0:t}^i = (o_0, \ldots, o_t)$, or implicit memory, on which it conditions its policy $\mu^i(a_t^i|o_{0:t}^i)$, to perform action selection. The environment then transitions to a new state in response to the joint action and current state, according to the state transition function $P(s_{t+1}|s_t, \mathbf{a}_t)$ and provides a shared numerical reward to each agent according to a reward function $r(s, a) : \mathcal{S} \times \mathcal{A} \to \mathbb{R}$. We define an agent's return as its discounted cumulative

---

*To be made available online after double-blind review process.

rewards over the $T$ episode timesteps, $G^i = \sum_{t=0}^{T} \gamma^t r_t^i$, where $\gamma \in (0, 1]$ is a scalar discount factor controlling how myopic agents are with respect to rewards received in the future. The goal of MARL in a Dec-POMDP is to find a joint policy $(\pi^i, \dots, \pi^n) \equiv \pi$ such that the return of each agent $i$, following $\pi^i$, is maximised with respect to the other agents' policies, $\pi^{-i} \equiv (\pi \backslash \pi^i)$. That is, we aim to find $\pi$ such that:

$$\forall i : \pi^i \in \arg\max_{\hat{\pi}^i} \mathbb{E}\left[ G^i \mid \hat{\pi}^i, \pi^{-i} \right]$$

## 2.2 Independent Q-Learning

The Q-value function $Q^\pi(s, a)$ for a policy $\pi(\cdot \mid s)$ is the expected sum of discounted rewards obtained by choosing action $a$ at state $s$ and following $\pi(\cdot \mid s)$ thereafter. DQN (Mnih et al., 2013) is an extension of Q-Learning (Watkins, 1989) which learns the Q-function, approximated by a neural network $Q_\theta$ with parameters $\theta$, and follows an $\epsilon$-greedy policy with respect to the learnt Q-function. One limitation of DQN is that it can only by applied to discrete action environments. DDPG (Lillicrap et al., 2016), on the other hand, is an extension of DQN which can be applied to continuous-action environments by learning a deterministic policy $\mu(s) : \mathcal{S} \to \mathcal{A}$ which is trained to output the action which maximises the Q-function at a given state.

Tampuu et al. (2015) showed that in a multi-agent setting such as *Pong*, independent DQN agents can successfully be trained to cooperate. Similarly, independent DDPG agents have successfully been trained in multi-agent environments (Lowe et al., 2017).

To train independent DDPG agents in a Dec-POMDP we instantiate a Q-function $Q_\theta^i(o_{0:t}^i, a_t^i)$ for each agent $i \in \mathcal{N}$, which conditions on each agents own observation history $o^i$ and action $a^t$. In addition, we also instantiate a policy network for each agent $\mu_\phi^i(o_t^i)$ which takes agent observations $o_t^i$ and maps them to actions $a_t^i$. Each agent's Q-function is independently trained to minimise the temporal difference (TD) loss, $\mathcal{L}_Q(\mathcal{D}^i)$, on transition tuples, $(o_t^i, a_t^i, r, o_{t+1}^i)$, sampled from its experience replay buffer $\mathcal{D}^i$ collected during training, with respect to parameters $\theta^i$:

$$\mathcal{L}_Q(\mathcal{D}^i, \theta^i) = \mathbb{E}_{o_t^i, a_t^i, r_t, o_{t+1}^i \sim \mathcal{D}} \left[ (Q_{\theta^i}^i(o^i, a^i) - r_t - \gamma \hat{Q}_{\theta^i}^i(o_{t+1}^i, \hat{\mu}_\phi^i(o_{t+1})))^2 \right]$$

where $\hat{Q}_\theta$ and $\hat{\mu}_\phi$ are delayed copies of the Q-network and policy network respectively, commonly referred to as the target networks. The policy network is trained to predict, given an observation $o^i$, the action $a^i$ that maximises the Q-function, which can be achieved by minimising the following policy loss with respect to parameters $\phi^i$:

$$\mathcal{L}_\mu(\mathcal{D}^i, \phi^i) = \mathbb{E}_{o_t^i \sim \mathcal{D}^i} \left[ -Q_{\theta^i}^i(o_t^i, \mu_{\phi^i}^i(o_t^i)) \right]$$

To improve the performance of independent learners in a Dec-POMDP, agents usually benefit from having memory (Hausknecht & Stone, 2015). Accordingly, we can condition the Q-networks and policies on observation histories $o_{0:t}^i$ instead of just individual observations $o_t^i$. In practice we use a recurrent layer in the neural networks. In addition, to further stabilize learning, we use eligibility traces (Sutton & Barto, 2018) in the form of $Q(\lambda)$, from Peng & Williams (1994).

## 3 Related Work

The concept of reusing computation for learning in some capacity is neither new, nor constrained to the domain of RL. We feel that topics such as transfer learning to new tasks (Bozinovski & Fulgosi, 1976), fine-tuning (e.g. Sharif Razavian et al. (2014)), and post-deployment model updates[*] fit into this broad philosophy. In RL specifically, the concept has also existed for some time (e.g. Fernández & Veloso (2006)), and other RL researchers are currently pursuing similar aims with different nomenclature (e.g. using offline RL as a 'launchpad' for online RL[†]). Indeed, Agarwal et al. (2022) accurately highlight that their conception of the field of reincarnation is a formalisation of that which already exists.

---

[*]See *Updatable Machine Learning* (UpML) workshop: https://upml2022.github.io/

[†]See *Offline RL* workshop: https://offline-rl-neurips.github.io/2022/

In MARL, too, there are extant works with the flavour of reincarnation. For example, both Kono et al. (2014) and Gao et al. (2021) explored the concept of 'knowledge reuse' in MARL. In a large-scale instance, Vinyals et al. (2019) naturally reused computation for the training of their AlphaStar system. Specifically, it is also interesting to note their concept of using agents to help train other agents with a 'league' algorithm. In a sense, this approach is somewhat similar to one of the anticipated benefits of selective reincarnation, where good agents can assist with teaching bad agents.

Nonetheless, we believe there has not yet been a formalisation of the field of multi-agent reincarnation, akin to the efforts done by Agarwal et al. (2022). Moreover, it seems that being selective in the agent reincarnation choice is also a novel specification.

## 4    DEFINITIONS

**Definition 1 (Multi-Agent Reincarnation)**  In a MARL system (see Section 2.1) with the set $\mathcal{N}$ of $n$ agents, an agent $i \in \mathcal{N}$ is said to be reincarnated (Agarwal et al., 2022) if it has access to some artefact from previous computation to help speed up training from scratch. Typically such an agent is called a *student* and the artefact from previous computation is called a *teacher*. The set of teacher artefacts in the system is denoted $T$. There are several types of artefacts which can be used as teachers, including (but not limited to): teacher policies $\pi_T$ or $\mu_T$, offline teacher datasets $\mathcal{D}_T$, and teacher model weights $\phi_T$ or $\theta_T$.

**Definition 2 (Selective Reincarnation)**  A selectively reincarnated MARL system with $n$ agents is one where $y \in [1, n)$ agents are trained from scratch (i.e. *tabula rasa*) and $x = n - y$ agents are reincarnated (Agarwal et al., 2022). The sets of reincarnated and tabula rasa agents are denoted $X$ and $Y$ respectively. A MARL system with $y = n$ is said to be fully tabula rasa, whereas a system with $x = n$ is said to be fully reincarnated.

## 5    CASE STUDY: SELECTIVELY-REINCARNATED POLICY-TO-VALUE MARL

Agarwal et al. (2022) presented a case study in *policy-to-value* RL (PVRL), where the goal is to accelerate training of a student agent given access to a sub-optimal teacher policy and some data from it. Similarly, we now present a case study in multi-agent PVRL, focusing on one of the methods invoked by Agarwal et al. (2022), called 'Rehearsal' (Gülçehre et al., 2020).

We set up our experiments as follows. We use an independent DDPG (Lillicrap et al., 2016) configuration, with some minor modifications to enable it to leverage offline teacher data for reincarnation. Specifically, we make two changes. Firstly, we compose each mini-batch of training data from 50% offline teacher data and 50% student replay data, similar to Gülçehre et al. (2020). This technique should give the student the benefit of seeing potentially high-reward transitions from the teacher, while also getting to see the consequences of its own actions from its replay data. Secondly, we add layer-norm to the critic network, to mitigate extrapolation error due to out-of-distribution actions, as per Ball et al. (2023).

For the sake of the current question of selective reincarnation, we use the HALFCHEETAH environment, first presented by Wawrzynski (2007), and later brought into the MuJuCo physics engine (Todorov et al., 2012). Specifically, we focus on the variant introduced by Peng et al. (2021) with their Multi-Agent MuJoCo (MAMuJoCo) framework, where each of the six degrees-of-freedom is controlled by a separate agent. We denote these six agents as the following: the back ankle (`BA`), the back knee (`BK`), the back hip (`BH`), the front ankle (`FA`), the front knee (`FK`), and the front hip (`FH`). This ordering corresponds to the array indices in the MAMuJoCo environment, from 0 to 5 respectively. We illustrate the HALFCHEETAH setup in the appendix, in Figure A.1.

For the set of proficient teacher policies, we initially train on the 6-agent HALFCHEETAH using tabula-rasa independent DDPG over 1 million training steps, and store the experiences using the `OG-MARL` framework (Formanek et al., 2023) so that they can be used as the teacher datasets. We then enumerate all combinations of agents for reincarnation, a total of $2^6 = 64$ subsets. With each subset, we retrain the system on HALFCHEETAH, where that particular group of agents gains access to their teachers offline data (i.e. they are reincarnated). For each combination, we train the system for $200k$ timesteps, remove the teacher data, and then train for a further $50k$ timesteps on student

data alone. Each experiment is repeated over five seeds. For the 'maximum return' metric, we find the timestep at which the return, averaged over the five seeds, is highest. For the 'average return' metric, we average the return over all seeds and all timesteps. We use these metrics as proxies for performance and speed to convergence respectively.

## 5.1 IMPACT OF TEACHER DATASET QUALITY

To begin with, we fully reincarnate the MARL system, giving all of the DDPG agents access to their teachers' datasets. Since the quality of the samples in the teacher's dataset likely has a marked impact on the learning process, we create two datasets for comparison: 'Good' and 'Good-Medium', where these names indicate the typical returns received across samples[*]. Figure A.2, in the appendix, shows the distribution of the returns in these two datasets.

We run the fully reincarnated configuration with each of these datasets, along with a *tabula rasa* baseline. Figure 1 presents these results.

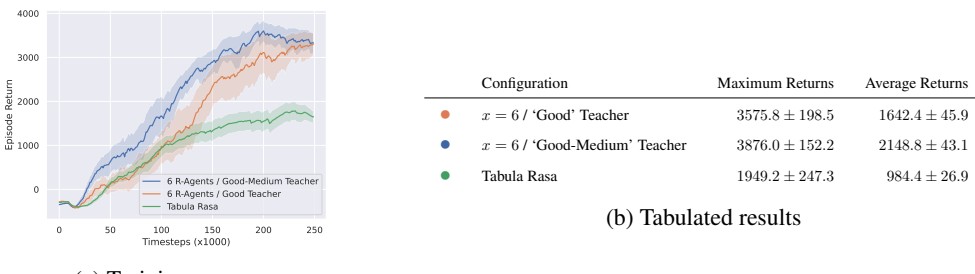

| | Configuration | Maximum Returns | Average Returns |
|---|---|---|---|
| 🔴 | $x = 6$ / 'Good' Teacher | $3575.8 \pm 198.5$ | $1642.4 \pm 45.9$ |
| 🔵 | $x = 6$ / 'Good-Medium' Teacher | $3876.0 \pm 152.2$ | $2148.8 \pm 43.1$ |
| 🟢 | Tabula Rasa | $1949.2 \pm 247.3$ | $984.4 \pm 26.9$ |

(b) Tabulated results

(a) Training curves

Figure 1: Performance using the two different teacher datasets. In the plot, a solid line indicates the mean value over the runs, and the shaded region indicates one standard error above and below the mean. In the table, values are given with one standard error.

Notice in Figure 1a that providing access solely to 'Good' teacher data initially does *not* help speed up training and even seems to hamper it. It is only after around $125k$ timesteps that we observe a dramatic peak in performance, thereafter significantly outperforming the *tabula rasa* system. In contrast, having additional 'Medium' samples enables higher returns from the beginning of training – converging faster than the solely 'Good' dataset.

One may be surprised by these results – that it takes the system some time to realise benefits from high-return teacher data. However, we postulate that when using the 'Good' dataset, the teacher data is narrowly focused around high-return strategies, yet the corresponding state and action distributions are likely very different to the students' own state and action distributions early in training. Consequently, the students struggle to leverage the teacher datasets until later in training, when the state-action distribution mismatch is minimised. This belief is evidenced by the results in Figure 1, and further supports the notion that the quality of the teachers' datasets has an impact on the outcomes of reincarnation. We feel this research direction is itself a promising one for future works, which we discuss in more detail in our roadmap, in Section 6. For the purposes of this investigation though, focusing on selective reincarnation and not dataset quality, we simply report the balance of our results using the 'Good-Medium' dataset. Nevertheless, for completeness, we run our experiments with both datasets, and provide these results publicly[†].

## 5.2 ARBITRARILY SELECTIVE REINCARNATION

We now focus on the core aspect of our investigation: selective reincarnation. Firstly, we approach the problem at a high-level by reincarnating $x$ of the $n$ agents and aggregating across all combinations for that $x$. That is, we do not study *which* agents are selectively reincarnated for a given $x$. For example, for $x = 2$, we reincarnate all pairs of agents in separate runs: $\{(\texttt{BA}, \texttt{BK}), (\texttt{BA}, \texttt{BH}), \dots\}$, and then

---

[*]'Good' is created with roughly the last 20% of the various teachers' experiences from training, and 'Good-Medium' with the last 40%.

[†]To be made available online after double-blind review process.

average those results. As an important point, notice that the count of combinations depends on $x$, calculated as $\binom{x}{n} = \frac{x!}{n!(x-n)!}$ – e.g. there is just one way to reincarnate all six of the agents, but there are twenty ways to reincarnate three of the six agents. Accordingly, we average over a different count of runs depending on $x$, which affects the magnitude of the standard-error metrics. We highlight this detail to warn against comparing the confidence values across these runs. The essence of these results, instead, is to show the mean performance curve.

The returns from these runs, computed over five seeds times $\binom{x}{6}$ combinations, is given in Figure 2, with both the graphical plot and the tabular values reported.

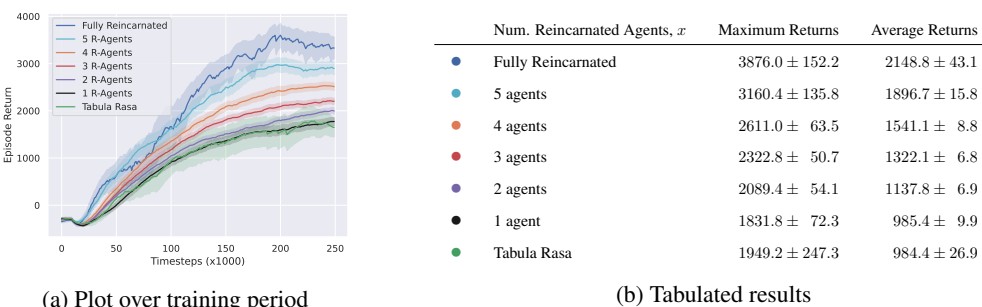

| (a) Plot over training period | | (b) Tabulated results |

| | Num. Reincarnated Agents, $x$ | Maximum Returns | Average Returns |
|---|---|---|---|
| ● | Fully Reincarnated | $3876.0 \pm 152.2$ | $2148.8 \pm 43.1$ |
| ● | 5 agents | $3160.4 \pm 135.8$ | $1896.7 \pm 15.8$ |
| ● | 4 agents | $2611.0 \pm \phantom{0}63.5$ | $1541.1 \pm \phantom{0}8.8$ |
| ● | 3 agents | $2322.8 \pm \phantom{0}50.7$ | $1322.1 \pm \phantom{0}6.8$ |
| ● | 2 agents | $2089.4 \pm \phantom{0}54.1$ | $1137.8 \pm \phantom{0}6.9$ |
| ● | 1 agent | $1831.8 \pm \phantom{0}72.3$ | $985.4 \pm \phantom{0}9.9$ |
| ● | Tabula Rasa | $1949.2 \pm 247.3$ | $984.4 \pm 26.9$ |

Figure 2: Selective reincarnation performance, aggregated over the number of agents reincarnated. In the plot, a solid line indicates the mean value over the runs, and the shaded region indicates one standard error above and below the mean. In the table, values are given with one standard error. A reminder: take caution when comparing the standard error metrics across values of $x$, since the number of runs depends on $\binom{x}{6}$.

In Figure 2a, we notice firstly that reincarnation enables higher returns. We already saw in Figure 1 that full reincarnation yields higher returns than *tabula rasa*, but we now see that a selectively-reincarnated setup also yields benefits – e.g. reincarnating with just half of the agents provides an improvement over *tabula rasa*. We do see that reincarnating with just one agent is somewhat detrimental in this case, with a slightly lower maximum return over the training period, but not significantly.

## 5.3 TARGETED SELECTIVE REINCARNATION MATTERS

Though the results from Figure 2 are interesting, we now present a vital consideration: in a multi-agent system, even in the simpler homogeneous case, agents can sometimes assume dissimilar roles (e.g. Wang et al. (2020) show the emergence of roles in various tasks). In the HALFCHEETAH environment particularly, consider the unique requirements for the ankle, knee, and hip joints, and how these differ across the front and back legs, in order for the cheetah to walk.

It is thus important that we compare, for a given $x$, the results across various combinations. That is, e.g., compare reincarnating (BA, BK) with (BA, BH), etc. Though we run experiments over *all* possible combinations, plotting these can quickly become unwieldly and difficult to study. Instead, we show here only the best and worst combinations for each $x$, as ranked by the average return achieved. These plots can be seen in Figure 3, with values tabulated in the appendix, in Table A.1. We release results for all combinations online[*].

We see in these graphs that the choice of which agents to reincarnate plays a significant role in the experiment's outcome. For example, consider the choice of reincarnating three agents, shown in Figure 3d: selecting (BH, FK, FH) instead of (BA, BK, FK) increases the maximum return by 33%, and almost doubles the average return. Similar improvements exist for other values of $x$.

We also notice an interesting pattern in the best subsets for reincarnation (denote the best subset for $x$ as $X_x^*$): as $x$ increases, agents are strictly added to the subset. That is, $X_1^* = \{\text{BH}\}$, $X_2^* = X_1^* \cup \{\text{FK}\}$, and so on. Moreover, for the best subset choices, the maximum returns monotonically increase with $x$,

---

[*]To be made available online after double-blind review process.

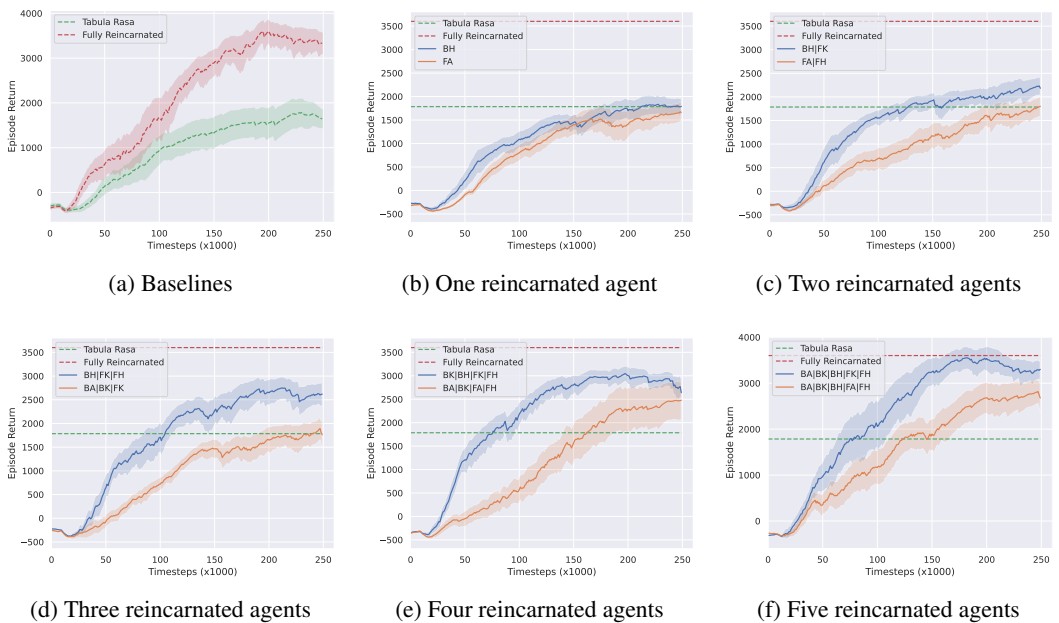

(a) Baselines      (b) One reincarnated agent      (c) Two reincarnated agents

(d) Three reincarnated agents      (e) Four reincarnated agents      (f) Five reincarnated agents

Figure 3: Training curves for the best and worst combinations of reincarnated agents, decided by the average episode return achieved. A solid line indicates the mean value over five seeds, and the shaded region indicates one standard error above and below the mean. In Figures 3b to 3f, the green and red lines indicate the maximum return achieved by the *tabula rasa* and fully-reincarnated approaches respectively.

up to full reincarnation. Interestingly, though, the average return (i.e. the time to convergence) is slightly higher for $x = 5$ than for full reincarnation, $x = n = 6$ (see Table A.1).

To affirm these points, we use the `MARL-eval` tool from Gorsane et al. (2022), built upon work by Agarwal et al. (2021), to plot the associated performance profiles and probability of improvement graphs in Figure 4, and the aggregate scores in Figure A.3.

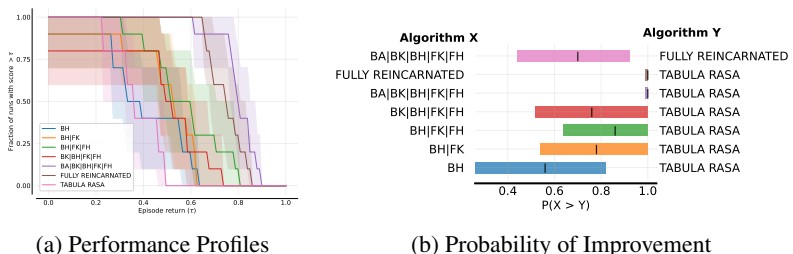

(a) Performance Profiles      (b) Probability of Improvement

Figure 4: `MARL-eval` (Gorsane et al., 2022; Agarwal et al., 2021) plots comparing the best performing combination of $x$ reincarnated agents for each $x \in [0, n]$ .

We use these results as clear evidence of the following: selective reincarnation can yield benefits, with higher returns and faster convergence over *tabula rasa* and possibly even full reincarnation; *but* one must be very careful of which agents are selected, for a bad choice can lead to a sub-optimal outcome.

Naturally, this diagnosis opens up many further questions. How can we know, ideally *a priori*, whether a given combination is a poor or excellent one? In this example of the HALFCHEETAH environment, we might try to reason about various combinations: e.g, from Figure 3f, we see that reincarnating the back leg, front hip, and *front knee* is a significantly better choice than the the back leg, the front hip, and the *front ankle* – does this result perhaps reveal something about the nature

of how HALFCHEETAH learns? We show some other interesting groupings in the appendix, in Figure A.4.

## 6    ROADMAP FOR MULTI-AGENT REINCARNATION

We now present a brief roadmap of some avenues to explore in this domain.

**Selective Reincarnation in MARL.** There are many other conceivable methods for doing selective reincarnation in MARL which we did not explore. In this work we focused on a method similar to 'rehearsal' (Gülçehre et al., 2020), but future works could experiment with methods such as 'jump-starting' (Uchendu et al., 2022), 'kick-starting' (Schmitt et al., 2018) and offline pre-training. We find offline pre-training a particularly promising direction for selectively reincarnating systems of independent DDPG agents – e.g. one could apply a behaviour cloning regularisation term to the policy loss in DDPG, as per Fujimoto & Gu (2021), and then to wean it off during training, as per Beeson & Montana (2022). Another direction could be to develop bespoke selective reincarnation methods; for example, a method to enable agents to 'trust' those agents with a teacher more than they would otherwise. Additionally, there is a trove of work to be done in how to understand which agents have the highest impact when reincarnated, and perhaps to reason about this delineation *a priori*. Finally, we also encourage larger-scale selective-reincarnation experiments on a wider variety of environments, and perhaps even tests with real-world systems.

**Beyond Independent Reincarnation.** In this paper, we focused on using independent DDPG for learning in MARL, but we believe many valuable open-problems exist outside of such an approach. For example, how does one effectively reincarnate MARL algorithms that belong to the paradigm of *Centralised Training Decentralised Execution* (CTDE), such as MADDPG (Lowe et al., 2017) and QMIX (Rashid et al., 2020)? It is not clear how one might selectively reincarnate agents with a centralised critic. In general, outside of just selective reincarnation, we also showed evidence that the quality of the teacher policy and data can have a large impact on the outcomes of reincarnation in RL. Exploring the benefits of, e.g., a curriculum-based, student-aware teacher could be an direction for future work. One could also explore ideas of curricula in the algorithm design itself – e.g. solely training the reincarnated agents' critics but freezing their policies, until the other agents 'catch up.' Another question we have for reincarnation in MARL is how teachers can help students learn to cooperate more quickly. Learning cooperative strategies in MARL can often take a lot of exploration and experience. Could reincarnating in MARL help reduce the computational burden of learning cooperative strategies from scratch? Many exciting avenues exist, and this work is only the beginning.

## 7    CONCLUSION

In this paper, we explored the topic of reincarnation (Agarwal et al., 2022), where prior computation is reused for future experiments, within the context of multi-agent reinforcement learning. Specifically, we proposed the idea of *selective* reincarnation for this domain, where not all the agents in the system are reincarnated. To motivate this idea, we presented a case study using the HALFCHEETAH environment, and found that selective reincarnation can result in higher returns than if all agents learned from scratch, and faster convergence than if all agents were reincarnated. However, we found that the choice of which agents to reincarnate played a significant role in the benefits observed, and we presented this point as the core takeaway. We used these results to argue that a fruitful field of work exists here, and finally listed some avenues that may be worth exploring, as a next step.

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

# A APPENDIX

## ENVIRONMENT: 6-AGENT HALFCHEETAH

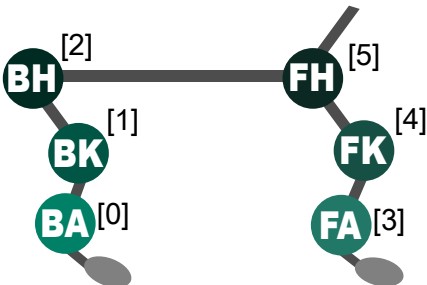

Figure A.1: The HALFCHEETAH environment (Wawrzynski, 2007; Todorov et al., 2012) viewed from the perspective of six separate agents (Peng et al., 2021). The array indices from the MAMuJoCo environment are given in brackets. Note that this diagram is purely illustrative and is not drawn with the correct relative scale.

## DATASET DISTRIBUTIONS

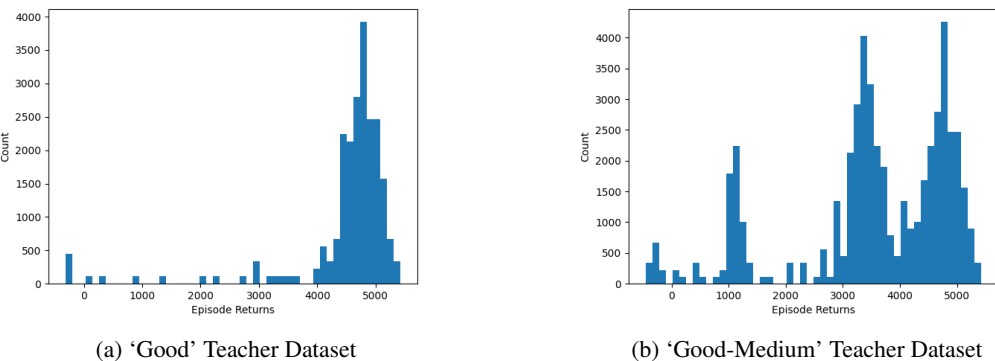

(a) 'Good' Teacher Dataset

(b) 'Good-Medium' Teacher Dataset

Figure A.2: Histograms of episode returns from the two different datasets.

TARGETED SELECTIVE REINCARNATION: BEST & WORST RESULTS

| | Configuration | Maximum Returns | Average Returns |
|---|---|---|---|
| ● | Tabula Rasa | $1949.2 \pm 247.3$ | $984.4 \pm 26.9$ |
| ● | BH | $1950.2 \pm 471.8$ | $1116.3 \pm 47.2$ |
| ● | FA | $1886.1 \pm 646.5$ | $900.1 \pm 48.0$ |
| ● | BH, FK | $2367.0 \pm 504.0$ | $1455.0 \pm 53.4$ |
| ● | FA, FH | $1990.7 \pm 472.8$ | $890.7 \pm 47.8$ |
| ● | BH, FK, FH | $2875.6 \pm 483.8$ | $1786.3 \pm 67.2$ |
| ● | BA, BK, FK | $2146.5 \pm 626.2$ | $961.2 \pm 50.4$ |
| ● | BH, FK, FH, BK | $3215.2 \pm 266.4$ | $2153.1 \pm 66.9$ |
| ● | BA, BK, FA, FH | $2610.2 \pm 1155.9$ | $1185.5 \pm 72.4$ |
| ● | BH, FK, FH, BK, BA | $3827.6 \pm 699.5$ | $2370.4 \pm 82.7\,^{*}$ |
| ● | BA, BK, FA, FH, BH | $2934.0 \pm 715.9$ | $1594.3 \pm 70.1$ |
| ● | Fully Reincarnated | $3876.0 \pm 152.2\,^{*}$ | $2148.8 \pm 43.1$ |

Table A.1: Return values for the best and worst runs for a given number of selectively reincarnated agents. An asterisk ($*$) indicates the highest value in each column. Values are given with one standard error.

RLIABLE AGGREGATE SCORES

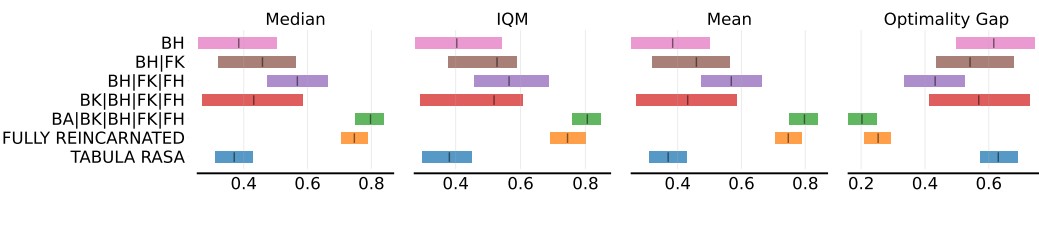

Figure A.3: `MARL-eval` (Gorsane et al., 2022; Agarwal et al., 2021) aggregate scores for each of the best performing combinations of $x$ reincarnated agents for each $x \in [0, n]$.

## SELECTIVE REINCARNATION ANECDOTES IN HALFCHEETAH

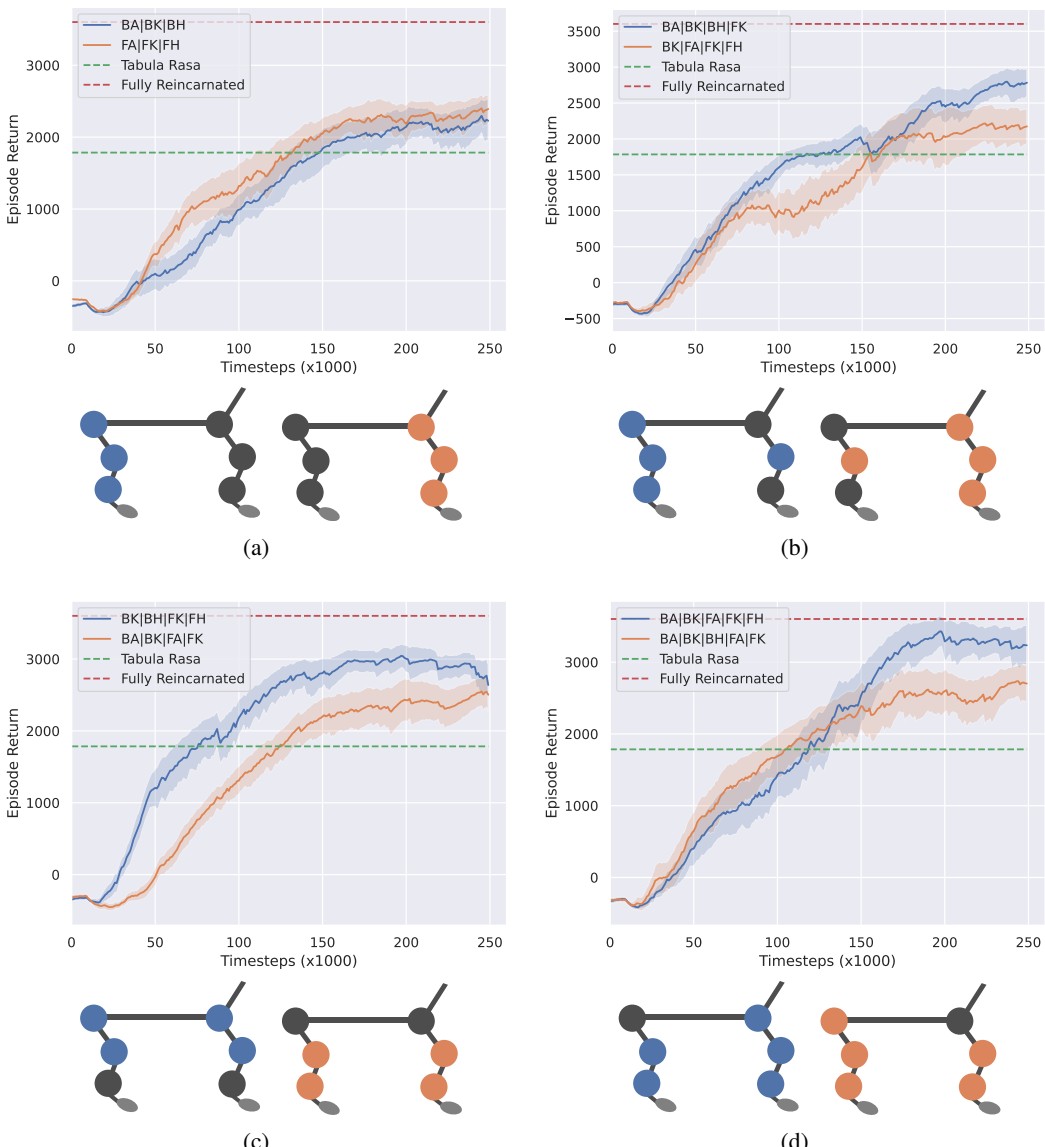

Figure A.4: Comparisons of some interesting selective reincarnation patterns in HALFCHEETAH. In the plots, a solid line indicates the mean value over the runs, and the shaded region indicates one standard error above and below the mean.

