# OpenReview forum: "Reduce, Reuse, Recycle: Selective Reincarnation in Multi-Agent Reinforcement Learning"
_ICLR.cc/2023/Workshop/RRL — RRL 2023 Oral_

### Official Review · Reviewer_yC6G · 2023-02-28

**Rating:** 4
**Confidence:** 5

**Review:**

The authors of this paper explored the use of prior computation in the context of multi-agent reinforcement learning through the lens of reincarnation. Specifically, they introduced the concept of selective reincarnation, where only certain agents within the system undergo reincarnation. In addition, the authors conducted ablation studies to investigate the impact of the number and selection of reincarnated agents. This research is a valuable contribution to the MARL community's interest in leveraging prior computations.

Here are a few comments/suggestions that might be helpful in improving the paper.
a- As mentioned by the authors and (Agarwal et al., 2022), using prior computations is not a novel idea. To improve the related work section, the authors could consider including other works in the context of MARL that use prior computations, such as using offline data [1] or pre-training self-play agents and fine-tuning them with novel partners [2].

b- Regarding the claim about faster convergence due to selective reincarnation, it may not be entirely clear from Figure 3, and the evidence is not sufficient. To support this claim, the authors could plot additional combinations of five agent reincarnations and compare them to the case of fully reincarnated agents. This would allow for a more robust analysis of the phenomenon.

[1] Meng et al, (2021) "Offline Pre-trained Multi-Agent Decision Transformer: One Big Sequence Model Tackles All SMAC Tasks"
[2] Nekoei et al, (2021) "Continuous Coordination As a Realistic Scenario for Lifelong Learning"

---

### Official Review · Reviewer_H3We · 2023-02-28
**proper extension of the reincarnation framework to MARL**

**Rating:** 4
**Confidence:** 4

**Review:**

This work extends the reincarnation framework to Multi-agent Reinforcement Learning (MARL).
It first defines Multi-agent Reincarnation and then Selective Reincarnation, in which only a subset of the agents are reincarnated.
Empirically, they show that arbitrary selective reincarnation interpolates the performance between Tabula Rasa and Full reincarnation as the subset of reincarnated agents increases.
Then, they show that targeted reincarnation can have a significant effect on performance and can sometimes offer small gains over full reincarnation.
This is a good paper and very much aligned with the spirit of the Workshop.
I'd encourage the authors to keep improving this work if a publication is a goal.
In its current state, there's not much insight to be gained from the experiments nor a new method that can be used in practice.

---

### Official Review · Reviewer_7nFZ · 2023-03-02
**review of Reduce, Reuse, Recycle: Selective Reincarnation in Multi-Agent Reinforcement Learning**

**Rating:** 3
**Confidence:** 4

**Review:**

The paper demonstrates that selectively re-using the policies of some agents are important to realize the benefit of Reincarnation in the multi-agent reinforcement learning setting. The setting studied is interesting and I agree with the paper that this is a rich area for further scientific exploration.